# Photosynthetic oxygen bubble stream sounds from aquatic macrophytes, and their consequences for acoustic biodiversity inventories and acoustic communication in shallow freshwater settings

**Katie Campbell**[ID]*, **Yelena Cerezke-Riemer, John H. Acorn**

Department of Renewable Resources, University of Alberta, Alberta, Canada

* kgc2@ualberta.ca

## Abstract

The emerging field of soundscape ecology focuses on biological, geophysical, and anthropogenic sounds, and provides a non-invasive method to inventory ecosystems. Most of the work on freshwater soundscapes focuses on larger fishes in deeper water, or on insects. We suggest the possibility that such studies have either missed or misidentified photosynthetic oxygen bubble sounds (POBS) produced by bubble streams from damaged macrophytes in sunny shallow water. These contribute significantly to local soundscapes. We recorded such sounds in the shallows of Gull Lake, Alberta, Canada, where POBS from sago pondweed (*Stuckenia pectinata*), along with water boatman stridulations (Hemiptera: Corixidae), comprised almost all of the sounds we encountered. These sounds attenuate rapidly with distance, and the POBS constitute a remarkable acoustic diversity, resulting in a patchwork of very different soundscapes in these shallows. Recognition of POBS has important consequences for acoustic bioinventories in shallow water, rapid ecosystem assessments involving indices of primary production, and bioacoustics studies of such organisms as corixid bugs, communicating against a cacophonous background of POBS.

## Introduction

Soundscape ecology, an emerging field, concerns itself with biological, geophysical, and anthropogenic sounds in nature [1,2], often with a focus on the impacts of anthropogenic sounds on vertebrate life [3,4]. Soundscapes create the acoustic backdrop for all organisms with a sense of hearing. The use of sound as an indirect, non-invasive method to assess aquatic biodiversity is also common practice [5–9]. However, the study of sound in freshwater is still developing [5], with very little known regarding the identification of particular sound sources [5–9].

In freshwater settings, studies have focused on sounds produced by fishes, amphibians, and arthropods [9], as well as sediment transport, organic decomposition [10], and human activities [10,11]. Plant sounds are less well understood [8] and most studies fail to mention sound production by plants during photosynthesis [5,9–12], with some exceptions [13–16].

**Data availability statement:** All relevant data (i.e., sound files) are fully available without restriction, as uploads to Xeno-canto, a public repository (https://xeno-canto.org/). The sound files referenced in the manuscript have the following Cat.nr.: XC920087, XC893189-XC893193. They are all found under the following user profile: https://xeno-canto.org/contributor/ZXWJFVRVKA?dir=0&order=xc

**Funding:** The author(s) received no specific funding for this work.

**Competing interests:** The authors have declared that no competing interests exist.

Sound production by marine algae on coral reefs [13,14] and by seagrass beds [17,18] are perhaps the best-known examples. Algae in coral reefs release oxygen bubbles, creating pinging [14] and ticking sounds [10,13]. Similar to algae, freshwater macrophytes release oxygen as gas bubbles, either through the leaf surface (a process that aquarists refer to as "pearling") [14] or as bubble streams from damaged tissues. Some aquatic beetles, known as plant breathers, pierce aquatic plants to replenish the oxygen in their plastron (physical gill) from the aerenchyma of the plant [19–21]. From damaged tissue, oxygen is able to escape, potentially contributing to what Rountree et al. refer to as a "cacophony of sounds" [12].

Bubble streams can be used to assess photosynthetic activity [9,10,13–15,22,23]. However, there has been a greater focus on controlled laboratory settings and the physics of regular bubble streams [15] than on the situation in the field. Lab settings allow for precise measurements of gas composition, bubble size, and the timing of bubble release. However, in natural environments, bubble production is subject to changing light regimes, wave action, damage to plant tissues [15], plant tissue elasticity, and other factors [24–28].

Determining the origins of aquatic sounds is a complicated process. Unknown sounds in freshwater are often attributed to fishes or insects [9–12], perhaps due to acoustic similarities to the sounds of known fishes, or those of terrestrial insects such as crickets and cicadas. Generally, however, soundscape studies are conducted in deeper water, where sport fish occur, often in sites too deep for rooted macrophytes. In shallower settings, the stridulations produced by aquatic insects have attracted some research attention, especially those of water boatmen (Hemiptera: Corixidae) [29–31]. Most aquatic insect sounds attenuate quickly over distance in water [32–34], making the notion of a soundscape highly localised for most organisms, especially shallow water organisms.

In the course of prospecting for undescribed insect sounds in shallow freshwater lakes in Alberta, Canada, we encountered additional sounds produced by aquatic macrophytes, and set about characterising these sounds and confirming their origins. Our results contrast somewhat with those of other published studies, with consequences for soundscape ecology in similar environments.

## Materials and methods

Recordings were made using an Aquarium Audio H2a-XLR hydrophone, a Sound Devices Mix Pre microphone preamp, and a Tascam Linear PCM Recorder/Mixer DR-60D with gain settings maximised on both the preamp and the sound recorder. The sampling rate was 44100 Hz. Spectrograms and waveforms were created from unprocessed selected exemplar recordings using Raven Lite2 software [35]. Spectrograms were produced with a short-time Fourier transform and a percent overlap (hop size) of 50%.

Recordings were made while wading in the shallows of Gull Lake. The hydrophone was handheld, and thus slight changes in its position affected the amplitude of recorded sounds, since these sounds attenuated markedly at small distances from the hydrophone. The hydrophone was typically held within the bottom 10 cm of the water column and suspended in a way that it was not touching plants or the substrate, to avoid friction sounds. Visual assessments were conducted during recordings via direct observation from above the water, through polarised sunglasses (the water is typically clear enough for easy viewing), as well as underwater with waterproof cameras, including a GoPro Hero, Safari 3 HD Action Camera, and a Panasonic Lumix TS20. The date, time, location, and in situ assessment of the apparent sound sources were noted as a verbal "slate" on the recording. Once we realised the possibility that most of the sounds were produced by bubble streams emanating from sago pondweed, we obtained simultaneous hydrophone and underwater video recordings (using a Lumix TS20 camera) of five sound-source sago pondweeds, and five silent sago pondweeds for

comparison. Recordings (both audio and video) were assessed in person; sounds were classified into broad categories. Five POBS were selected to illustrate the diversity we encountered. For these exemplars, the peak frequency, duration of repeated sections, and other relevant qualities of the sounds were characterised. These were manually determined in Raven Lite2 based on the peak frequency range of sounds on the spectrogram. Terms suggested by Baker and Chesmore [36] were used to characterise the sound. For example, a syllable is a single burst of sound that is separate from other syllables.

The primary recordings for this study were made in the sandy shallows of Gull Lake, Alberta, Canada, adjacent to the Summer Village of Gull Lake, in water 0.3 m deep or less. Recording took place during the daytime, under sunny conditions when macrophytes were actively photosynthesising, beginning between the hours of 10:00 and 14:00 in June, July, and August of 2021 and 2023.

Additional recordings were obtained at other shallow, freshwater lakes and slow-moving waters in Alberta (e.g., Open Creek Reservoir, Blindman River near Bentley, Guinevere Park (Edmonton), and Wabamun Lake).

Recordings were also made in aquaria indoors. A "50-gallon" (c. 32 cm X 121 cm basal dimensions X 50 cm high) glass aquarium with c. 20 cm water and 5+ cm of sand substrate, sloped up towards the sides, was used to record corixid stridulations. A smaller "five-gallon" glass aquarium (c. 21 cm X 40.5 cm basal dimensions X 27 cm high) with c. 24 cm water depth and a c. 3 cm sand substrate was used for sago pondweed aquarium recordings. The latter was illuminated by an overhead halogen desk lamp and light from a north-facing window c. 15 cm away.

Two gas samples, c. 1.0 ml each, were collected from sago pondweed bubble streams in Gull Lake, and in the five-gallon aquarium. Gas chromatography with thermal conductivity detection (GC-TCD), was used to quantify oxygen ($O_2$) concentrations. This method provides a reasonably accurate estimate of $O_2$ content, particularly when $O_2$ is present at or near 100%, due to TCD's sensitivity to thermal conductivity differences between gases. However, other potential gases in the sample, such as carbon dioxide (CO2) and nitrogen (N2) may lie below the detection limits of the method (100–1000 ppm).

## Results

In total, we obtained and examined 309 hydrophone recordings, including 98 from the sandy shallows of Gull Lake (Photo 1). The Gull Lake recordings cover more than 160 minutes and in addition to the focal sounds they also record our own vocal slating, wave sounds and sounds of shifting sands (on windy days), and the sounds of large gas bubbles emanating from the sediments in pocket (pock) marks kept free of vegetation by occasional gas upwellings. We interpreted the latter as methane craters, although we did not analyse the gasses produced, and more were less than one metre in diameter.

The clear shallow waters on the south end of Gull Lake made our work relatively straightforward, since direct observation of potential sound sources was possible. In terms of biological sounds, we determined that the majority of sounds on our recordings were produced by bubble streams emanating from the tissues of sago pondweed plants (*Stuckenia pectinata*).

In ten simultaneous hydrophone and underwater video recordings, bubble streams were visible from all five of the sound producing plants, and on none of the five soundless plants. The only species of macrophyte apparent to us in the Gull Lake sites was sago pondweed, a common plant in lakes and ponds in the region. Once we knew what to look for, we were able to identify a bubble stream coming from a sago pondweed plant each time we heard a POBS on the hydrophone. We also recorded the sound of one bubble-stream-producing sago

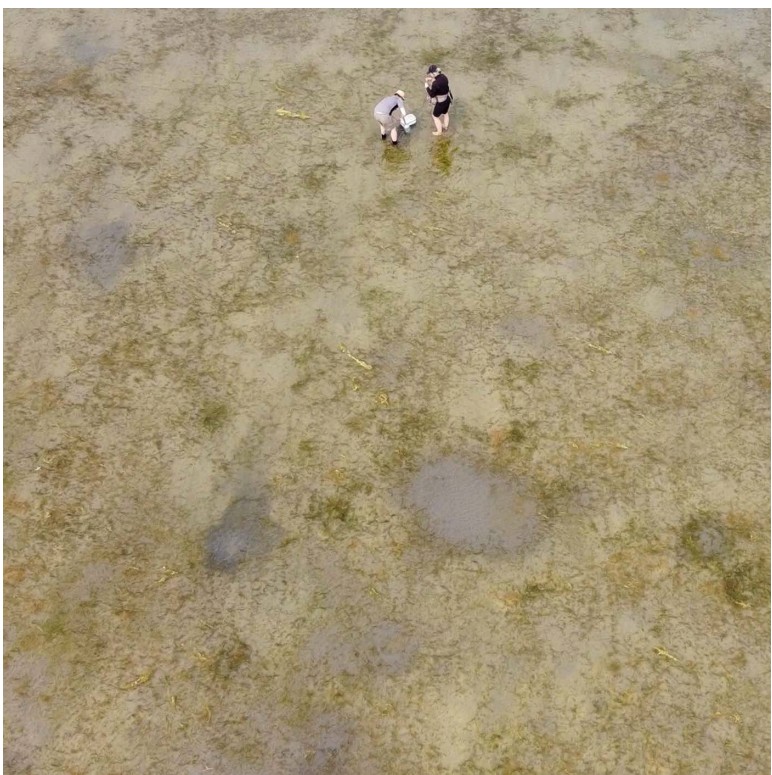

**Photo 1. Aerial (drone) view of Gull Lake shallows, with two researchers for scale.** The fine dark reticulations are sections of sago pondweed plants, and a number of methane craters are visible as well.

pondweed plant in an aquarium (Photo 2), and that sound was similar to Exemplar 3 below in most ways, including range of peak frequencies. At locations other than Gull Lake, in the presence of macrophytes, we encountered sounds that were qualitatively similar to the sago pondweed POBS, strongly suggesting a shared origin. However, we did not investigate these other sounds further.

Gas samples from a bubble stream collected from a plant in the lake, and from the aquarium plant, were found to contain c. 100% oxygen, as expected from plant tissues during sunlit conditions. We therefore interpreted the focal sounds as photosynthetic oxygen bubble sounds, for which we suggest the acronym POBS.

Photosynthetic bubble stream sounds and corixid stridulations dominated our recordings, accounting for over 90% of the sounds we recorded. Three sound-producing species of corixids were present at Gull Lake (*Cenocorixa dakotensis*, *Cenocorixa bifida*, and *Corisella tarsalis*) and all were identifiable by their calls. Our identifications were confirmed in aquaria as well as in the field. Corixids were the most abundant aquatic insects in Gull Lake, but other potential sound makers (e.g., some dytiscid beetles) were also present in smaller numbers. Worldwide, more than 7000 aquatic insect species are thought to produce sounds [37], with sound producing taxa also listed by Aiken [29] and corixids with stridulator structures listed by Hungerford [38].

Aidan Sheppard [unpublished data], in a 2018 study of aquatic hemipterans and beetles in the same location at Gull Lake, dip-netted 380 sound-producing corixids, 233 non-sound producing corixids and other hemipterans, and 27 aquatic beetles. Other aquatic invertebrates encountered frequently during our study included mayfly larvae (Ephemeroptera), midge

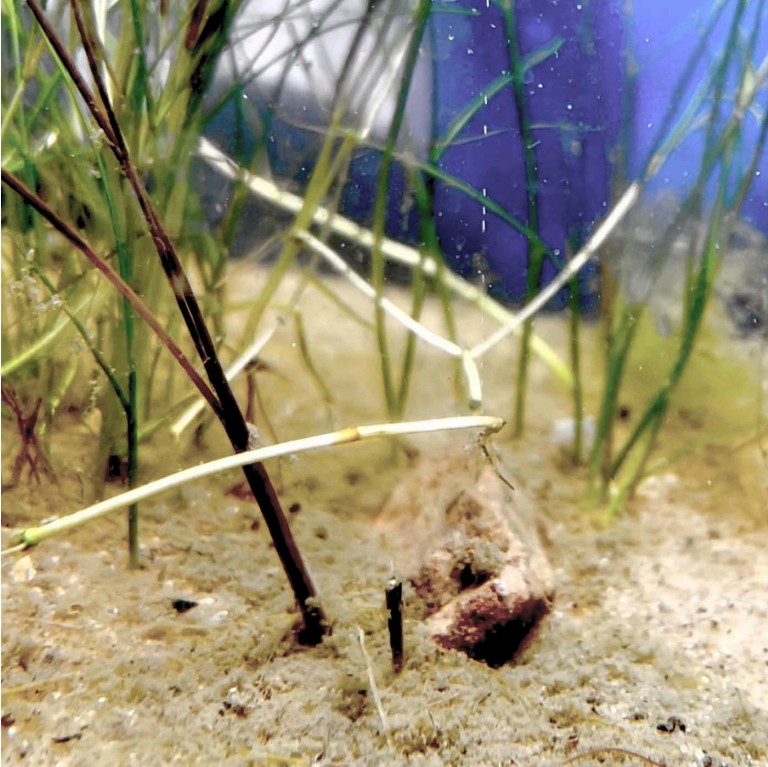

**Photo 2. Bubble stream emanating from a bend at the base of a sago pondweed stem.** Photographed in a five-gallon aquarium.

larvae (Diptera: Chironomidae), and scuds (Amphipoda: *Gammarus lacustris*), none of which are known or suspected to produce sounds.

Fishes in the Gull Lake shallows comprised only emerald shiners (*Notropis atherinoides*) and brook stickleback (*Culaea inconstans*); both present in small numbers, but apparently not producing sound despite us closely following numerous fishes with the hydrophone.

We did not record any sounds originating from fishes, animals other than corixids (including swimming and wading birds), or anthropogenic sounds such as boat motors. Boat motor sounds did not propagate into the shallow water even when boats were clearly visible, perhaps because of the dampening effect of the sand substrate, and the wide shallow zone of the lake.

The POBS sounds produced by sago pondweed have not been described before, and we therefore undertake to characterise them here. The five recordings we chose as exemplars for the diversity of photosynthetic oxygen bubble stream sounds from sago pondweed are characterised as follows, with numerical data presented in Table 1.

### Exemplar 1

Recorded on July 22, 2023, the first sound (Fig 1) was captured in shallow (<20 cm) water. A bubble stream, visible from above, was the source of the sound produced. Mean syllable length was 221.64 ms, with peak frequencies ranging from 1.6 to 2.6 kHz. Duration and amplitude of each syllable were consistent for the first four seconds before becoming more muted and less easily distinguished. Frequency also decreased during this period. Audibly, the sound is a periodic "quack quack quack" oscillation, slowing down during the second half of the recording.

**Table 1. Spread of peak frequency (kilohertz) and average duration of a single syllable (milliseconds) for five exemplar recordings of photosynthetic oxygen bubble sounds produced by sago pondweed (*Stuckenia pectinata*). Standard deviation and standard error of the mean are included for both frequency range and duration of syllables. The upper and lower standard deviation and standard error refer to the variability of either end of the mean range given. Values of N/A were unable to be provided as there is only one syllable present (Exemplar 5).**

| | Spread of Peak Frequencies (kHz) | | | Duration of Syllables (ms) | | |
|---|---|---|---|---|---|---|
| | Mean | SD | SE | Mean | SD | SE |
| Exemplar 1 | 1.6–2.6 | Lower: 0.35 Upper: 0.40 | Lower: 0.06 Upper: 0.70 | 221.64 | 30.99 | 5.39 |
| Exemplar 2 | 1.6–3.5; | Lower: 0.25 Upper: 0.43 | Lower: 0.04 Upper: 0.06 | 951.61 | 162.41 | 23.69 |
| | 5.3–7.5 | Lower: 0.33 Upper: 0.21 | Lower: 0.07 Upper: 0.02 | | | |
| Exemplar 3 | 0.9–6.2 | Lower: 0.15 Upper: 0.14 | Lower: 0.02 Upper: 0.02 | 10.22 | 2.40 | 0.34 |
| Exemplar 4 | 1.4–1.6; | Lower: 0.72 Upper: 0.70 | Lower: 0.02 Upper: 0.02 | 279.45 | 44.92 | 13.54 |
| | 2.9–3.1; | Lower: 0.12 Upper: 0.11 | Lower: 0.04 Upper: 0.03 | | | |
| | 4.5–4.7; | Lower: 0.16 Upper: 0.18 | Lower: 0.05 Upper: 0.05 | | | |
| | 6.0–6.2; | Lower: 0.22 Upper: 0.19 | Lower: 0.07 Upper: 0.06 | | | |
| | 7.6–7.8; | Lower: 0.28 Upper: 0.28 | Lower: 0.09 Upper: 0.08 | | | |
| | 9.0–9.4; | Lower: 0.34 Upper: 0.34 | Lower: 0.10 Upper: 0.10 | | | |
| | 10.9–11.0; | Lower: 0.37 Upper: 0.38 | Lower: 0.11 Upper: 0.11 | | | |
| | 12.5–12.7 | Lower: 0.45 Upper: 0.01 | Lower: 0.13 Upper: 0.01 | | | |
| Exemplar 5 | 1.6–4.7 | N/A | N/A | 4833.90 | N/A | N/A |

### Exemplar 2

Recorded on June 20, 2021, this sound was associated with a macrophyte bubble stream, visible from above in shallow (<20 cm) water. This sound ceased with removal of the plant. The sound varied considerably over time, but with an underlying regular oscillation (Fig 2). Long syllables averaged 951.61 ms, with two peak frequencies (1.6–3.5 kHz and 5.3–7.5 kHz). Amplitude peaked numerous times, most prominently in the middle of the recording. Audibly, this sound consists of repeated down-slurred "EEeeeuzzz" varying in amplitude over time.

### Exemplar 3

Recorded on August 1, 2021, in shallow (<20 cm) water in the morning (Fig 3), associated with a bubble stream captured on a waterproof camera. Numerous short syllables averaged 10.11 ms in duration, strung together as an oscillation with more-or-less consistent spacing and amplitude. Peak frequencies ranged from 0.9 to 6.2 kHz. Qualitatively, the short bursts are strung together to produce a whirring sound, overlayed by a lower, more constant buzz.

### Exemplar 4

Recorded August 1, 2021, Alberta, in shallow (<20 cm) water, associated with a bubble stream captured on a waterproof camera, this sound differed substantially from the others (Fig 4).

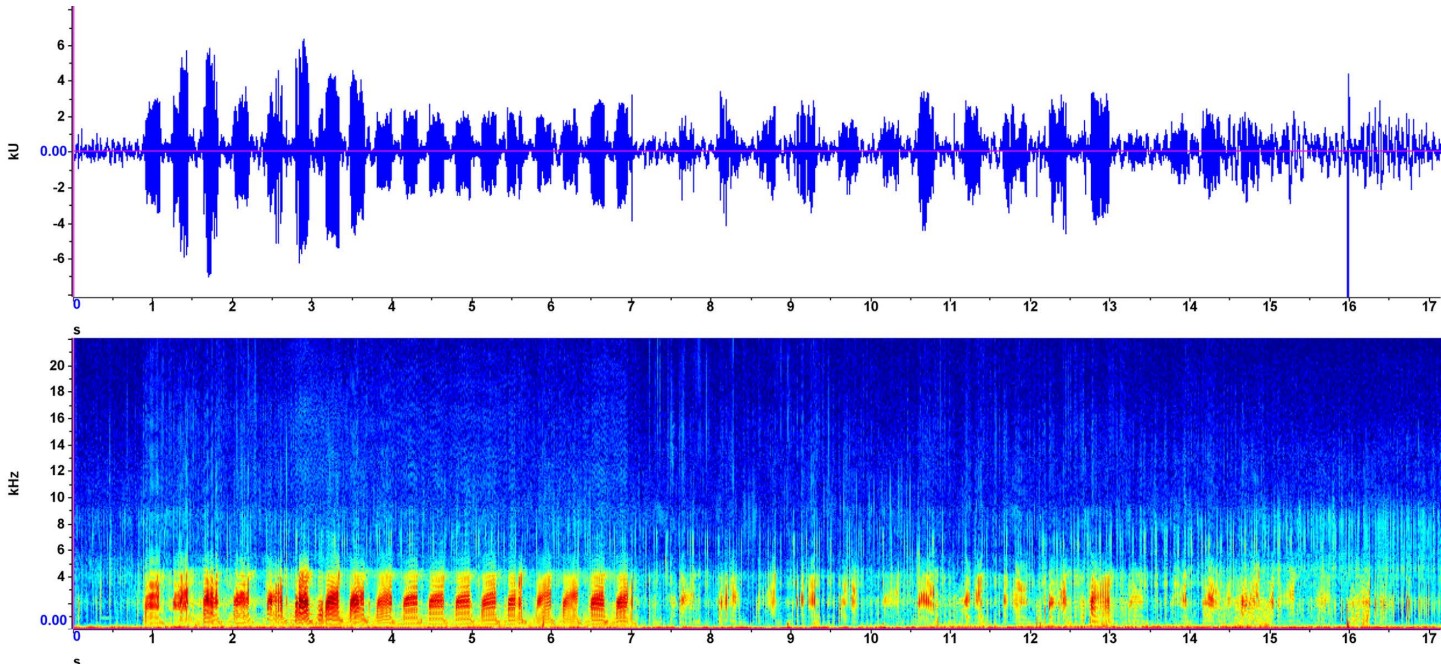

**Fig 1. Exemplar 1 spectrogram and waveform.** Recorded in Gull Lake, Alberta, Canada, July 22, 2023. Waveform view (top) shows time in seconds on the x-axis, and amplitude in kilounits (kU) on the y-axis. Spectrogram view (bottom) shows time in seconds on the x-axis, frequency in kilohertz (kHz) on the y-axis, and intensity represented by colour (blue = low intensity; yellow = medium; red = high). A short-time Fourier transform size with a 50% overlap parameter was used.

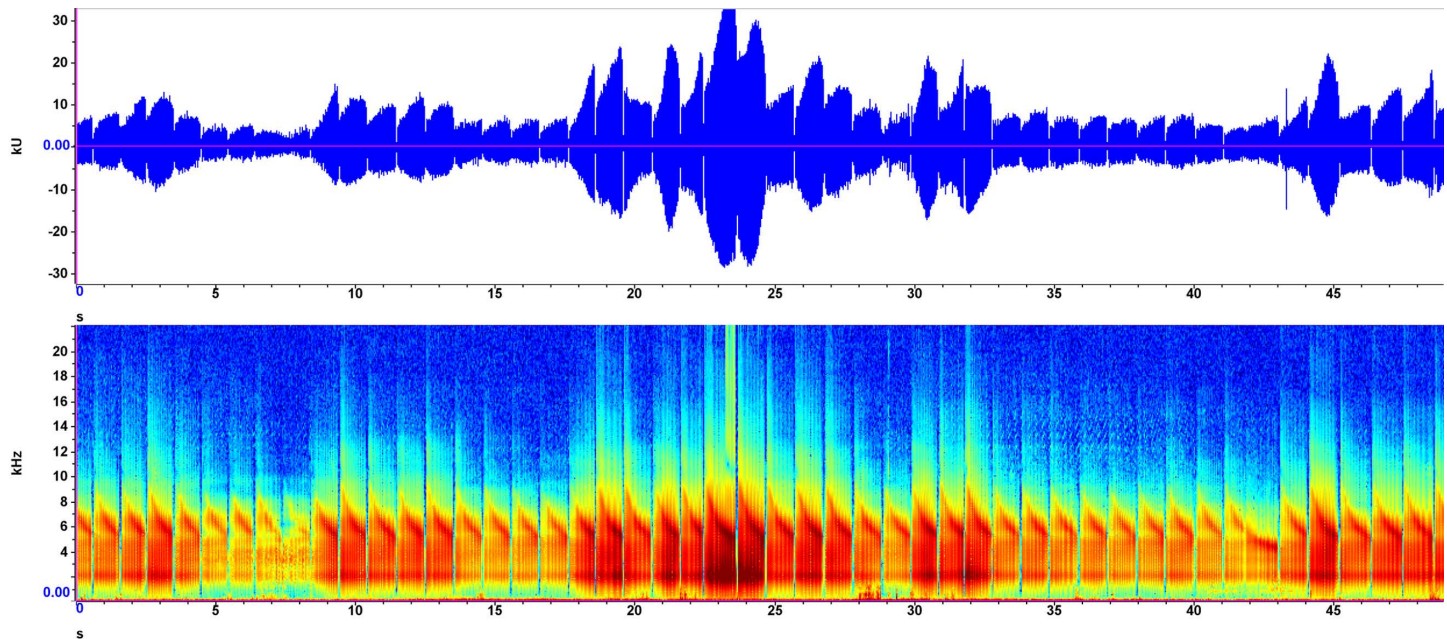

**Fig 2. Exemplar 2 spectrogram and waveform.** A macrophyte bubble stream sound recorded at Gull Lake, Alberta, Canada, June 20, 2021. See Fig 1 for axis descriptions.

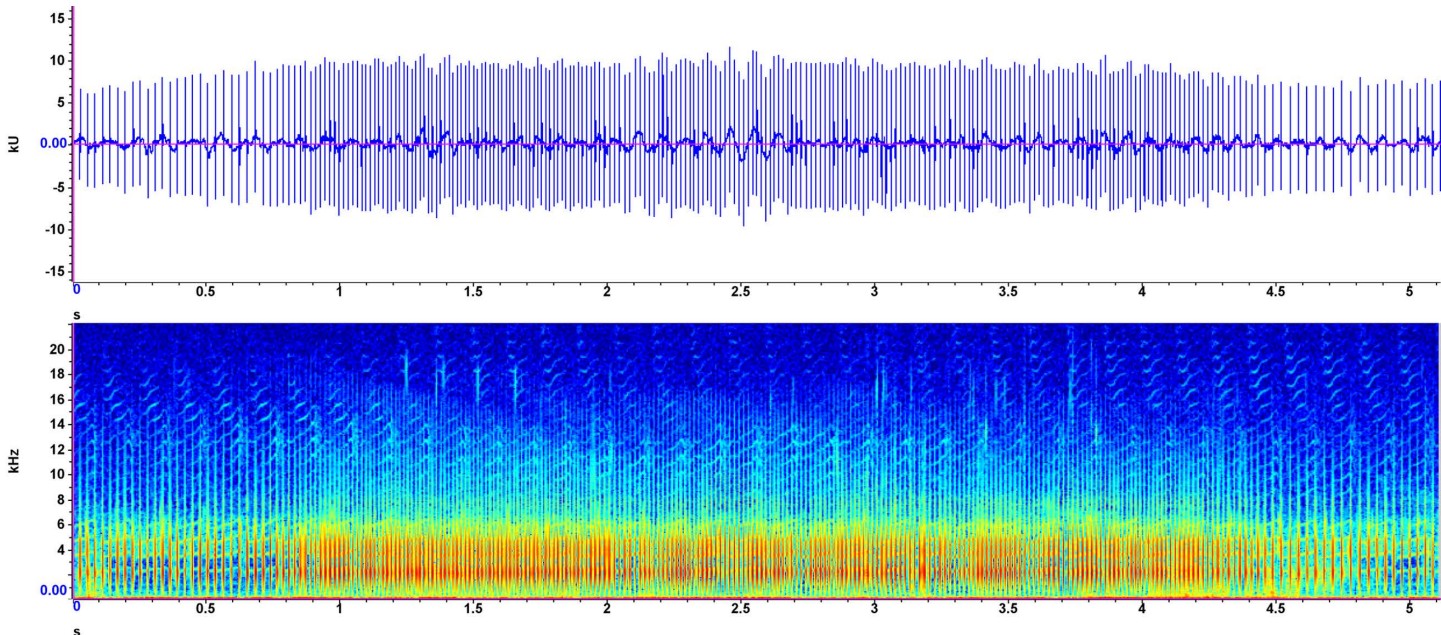

**Fig 3. Exemplar 3 spectrogram and waveform.** A macrophyte bubble stream sound recorded August 1, 2023, at Gull Lake, Alberta, Canada. See Fig 1 for axis descriptions.

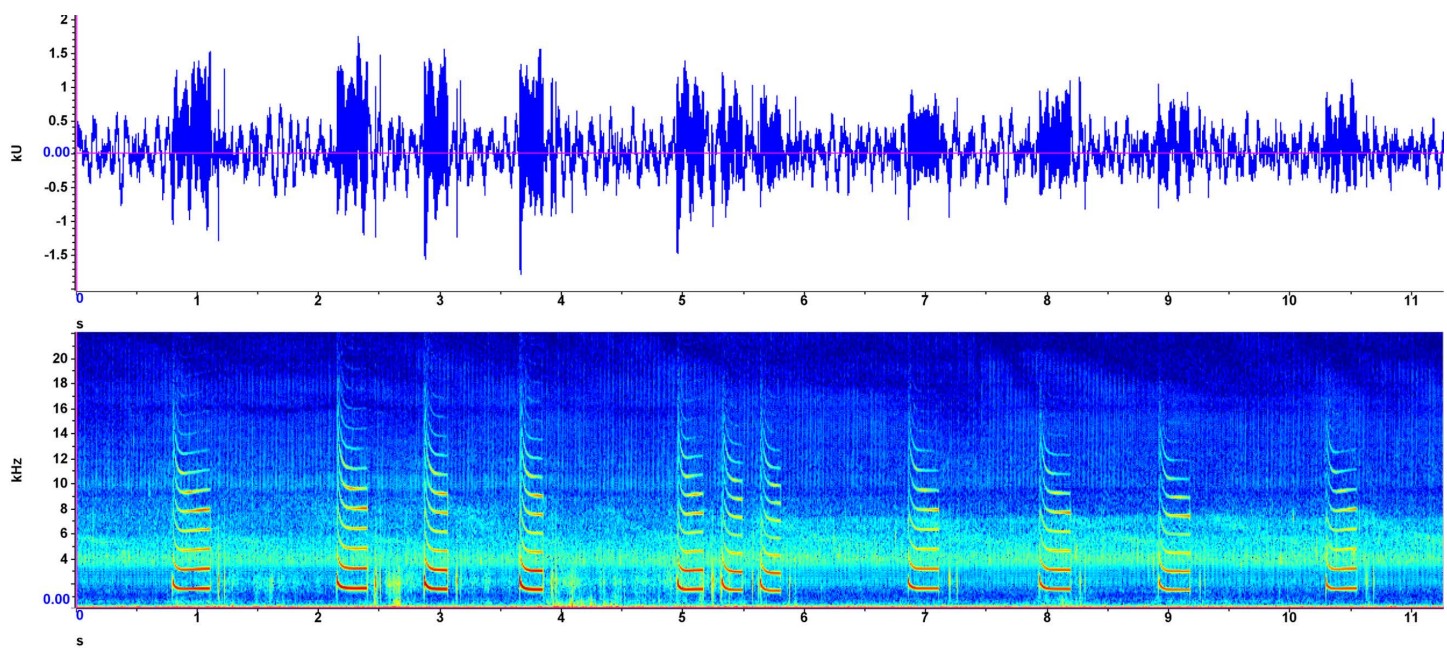

**Fig 4. Exemplar 4 spectrogram and waveform.** A macrophyte bubble stream sound recorded August 1, 2021, at Gull Lake, Alberta, Canada. See Fig 1 for axis descriptions.

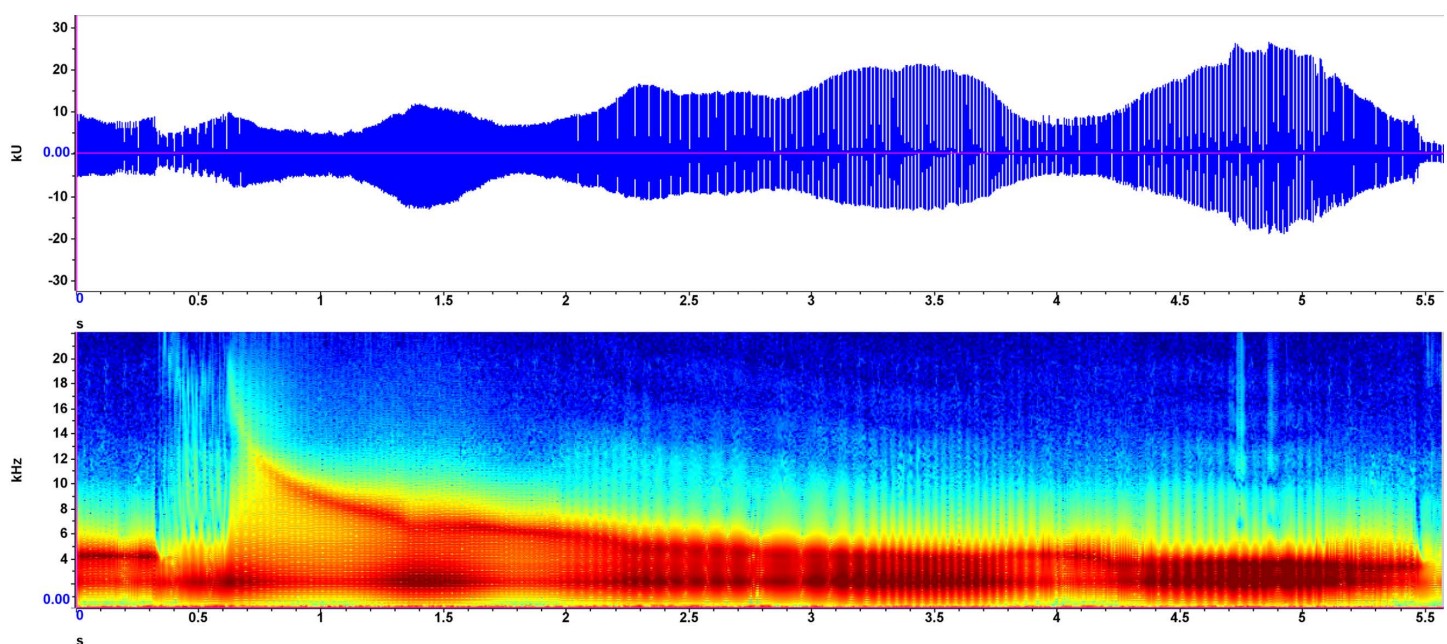

**Fig 5. Exemplar 5 spectrogram and waveform.** A macrophyte bubble stream sound recorded July 19, 2021, at Gull Lake, Alberta. See Fig 1 for axis descriptions.

Each syllable was, on average, 279.45 ms in duration, with 8 or more well-defined harmonics (peak frequencies listed in Table 1). The lower fundamental harmonics (below 12 kHz) were the most intense and had the greatest duration. Audibly, this sound featured irregularly spaced, high-pitched, "phwui" sounds, presumably caused by the build-up and release of pressure.

## Exemplar 5

Recorded on July 19, 2021, in shallow (<20 cm) water, associated with a bubble stream visible from above. This was a prolonged, continuous sound, lasting over 4000 ms, with peak frequencies ranging from 1.6 to 4.7 kHz (Fig 5). Frequency exhibited an initial sharp increase, gradually decreasing to consistent frequencies after two seconds. Audibly, it consists of a single "bweezzz" that decreases in amplitude over the duration of the recording.

These exemplar recordings were uploaded to the Xeno-Canto platform (https://xeno-canto.org, XC893189-XC893193), and the Freshwater Sounds Archive (https://fishsounds.net/freshwater.js), along with the aquarium-based sago pondweed recording (XC920087), by Katie Campbell.

## Discussion

The shallow water soundscape of Gull Lake, Alberta, was dominated by biological sounds produced by photosynthetic oxygen bubble streams from sago pondweed, along with occasional stridulations from water boatmen. Since the POBS were associated with single streams of bubbles (not "pearling" on the plant surface), we believe that wounds were the source of the bubbles. These were likely caused by aquatic insects, water birds, or wave action.

Prolonged oscillations were typical of the POBS, as expected from bubbles escaping through a small, elastic opening, via the buildup and release of pressure. The five sound exemplars presented here should not be interpreted as representing five discrete categories of

sounds. We suspect that the full range of sounds produced by POBS form a continuum, such that novel sounds will frequently be encountered.

Our visits to other freshwater lakes and ponds in central Alberta revealed similarly diverse soundscapes wherever we placed the hydrophone among rooted macrophytes in clear water during sunny conditions. In all cases, we found that the soundscapes in these environments were dominated by POBS, with corixid stridulations as the only other biological components, although in some settings the vocalisations and splashings of nearby birds (e.g., American Coot (*Fulica americana*), and Canada Goose (*Branta canadensis*)) were audible through the hydrophone.

It is well understood that freshwater sounds can originate from various sources, including fishes, amphibians, insects, macrophytes, and hydrological/geological processes [5,9–14]. In the freshwater aquatic literature, biological aquatic sounds are typically classified as either fast repetitive ticks (FRT) produced by air released from the swim bladders of fishes [39], or as stridulations produced by insects [12,39,40]. Since both POBS and FRTs involve bubbles escaping through an opening under pressure, we should expect POBS and FRTs to be similar. However, FRTs are shorter in duration, and occur more frequently at night due to nocturnal gas venting during the diel movements of fishes moving closer to the water surface [39]. It also appears that FRTs are more characteristic of deeper water, unlike macrophytic bubble sounds [27].

Stridulatory communication has been well documented in the Corixidae [29–31]; however, corixid stridulations are both quantitatively and qualitatively easy to distinguish from POBS (see the Xeno-Canto platform for exemplars of corixid stridulations, https://xeno-canto.org). Many unknown freshwater aquatic sounds have often been attributed to insects [12], but we note that many POBS are similar to terrestrial insect sounds, and especially those of some cicadas. This similarity, however, may be coincidental, and may mislead researchers when characterising aquatic sounds. Since we did not hear stridulations other than three familiar corixid species, we suspect that other putatively sound-producing organisms at our study sites either did not produce sound at all, or that they only stridulate occasionally as an anti-predator defense, as do many terrestrial beetles.

We therefore recommend caution when assuming that unidentified/unconfirmed aquatic sounds are produced by fishes or insects [12,16,40]. We believe that POBS are underreported in the current literature. We recognise the call from Rountree et al. [40] to further classify fish sounds in aquatic soundscapes, but expand it to encompass all aquatic sounds, including POBS. Only one paper to our knowledge has explicitly mentioned freshwater macrophytes as a major component of shallow, freshwater soundscapes [15].

We also suggest considering how differences in soundscapes likely correlate with lake zonation (e.g., littoral, limnetic, profundal, benthic) and water parameters (e.g., salinity, pH, depth, trophic state, sediments). Because all of the POBS we heard were produced by sago pondweed, the diversity of potential sounds from this species is broad. This macrophyte is found globally in freshwater habitats <2.5 metres deep with currents <1 m/s [41]. Given its ubiquity, similar soundscapes are likely to exist in shallow freshwater lakes elsewhere. However, many other species of macrophytes are undoubtedly capable of producing a very similar range of sounds.

Recognizing the ubiquity of POBS in shallow freshwater has implications for other studies as well. For acoustic bioinventory work, the diversity of plant-produced sounds has obvious consequences for the characterisation of sound sources in shallow freshwater. Understanding plant sounds will enhance the capacity for ecological conditions of ecosystems to be measured, primary production to be measured, pollution events to be characterised, and future initiatives as well. For behavioural studies of acoustic communication in shallow freshwater environments, it is important to recognize that POBS produce the acoustic background against which

animal communication occurs, at least during sunny daylight conditions. Whether POBS impede communication by organisms, such as corixids, or whether they provide a potential acoustic refuge from predators [42,43] remains an intriguing question for further research.

## Acknowledgments

We thank Murray Gingras for gas analysis and help in the field early on in the project, Jesse Acorn for help in the field, and the members of the "Frost Lab" in the Department of Renewable Resources at the University of Alberta for helpful discussions of an earlier draft of this paper.

## Author contributions

**Conceptualization:** Katie Campbell, Yelena Cerezke-Riemer, John H. Acorn.

**Data curation:** Katie Campbell.

**Investigation:** Katie Campbell, Yelena Cerezke-Riemer, John H. Acorn.

**Project administration:** John H. Acorn.

**Resources:** John H. Acorn.

**Supervision:** John H. Acorn.

**Writing – original draft:** Katie Campbell.

**Writing – review & editing:** Katie Campbell, John H. Acorn.

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
