## [Decision Letter · Decision Letter 0]

31 Oct 2024

PONE-D-24-33566Photosynthetic oxygen bubble stream sounds from aquatic macrophytes, and their consequences for acoustic biodiversity inventories and acoustic communication in shallow fresh waterPLOS ONE

Dear Dr. Campbell,

Thank you for submitting your manuscript to PLOS ONE. After careful consideration, we feel that it has merit but does not fully meet PLOS ONE’s publication criteria as it currently stands. Therefore, we invite you to submit a revised version of the manuscript that addresses the points raised during the review process.

We look forward to receiving your revised manuscript.

Kind regards,

Charles William Martin

Academic Editor

PLOS ONE

Journal Requirements:

3. Thank you for stating the following in your Competing Interests section: “NO”

4. We note that your Data Availability Statement is currently as follows: 

“All relevant data are within the manuscript and its Supporting Information files.”

5. Please amend the manuscript submission data (via Edit Submission) to include author “Yelena Cerezke-Riemer”. 

6. Please ensure that you refer to Figure 5 in your text as, if accepted, production will need this reference to link the reader to the figure.

Reviewers' comments:

Reviewer's Responses to Questions

**Comments to the Author**

1. Is the manuscript technically sound, and do the data support the conclusions?

Reviewer #1: Yes

Reviewer #2: Yes

Reviewer #3: Partly

2. Has the statistical analysis been performed appropriately and rigorously? 

Reviewer #1: N/A

Reviewer #2: Yes

Reviewer #3: N/A

3. Have the authors made all data underlying the findings in their manuscript fully available?

Reviewer #1: Yes

Reviewer #2: Yes

Reviewer #3: Yes

4. Is the manuscript presented in an intelligible fashion and written in standard English?

Reviewer #1: Yes

Reviewer #2: Yes

Reviewer #3: Yes

5. Review Comments to the Author

Reviewer #1: This manuscript describes the characteristics and occurrence of photosynthetic bubble stream sounds (POBS) from Sago pondweed as well as other sounds recorded in Gull Lake, Alberta, Canada. The authors provide visualizations and descriptions of five exemplar POBS recordings and discuss their attempts to identify or localize other sound sources in their study system. This type of observational or naturalist study provides valuable insights needed to untangle the diversity of sounds found in aquatic soundscapes as well as support future efforts to apply passive acoustic monitoring to what are often understudied underwater ecosystems. I found this study particularly interesting because it focuses on a less discussed category of biological sounds.

I appreciated the study and its efforts and I think it will be a valuable contribution to the scientific literature. The manuscript’s contents fall within the journal’s scope (i.e., by conducting original research contributing to our knowledge of underwater sound sources). As far as I could ascertain, the methods employed by the authors were valid and appropriate for their study objectives and their conclusions were generally reasonable and relevant to the work conducted. I especially appreciated their provision of multiple, diverse exemplar recordings with associated descriptions and measurements.

Though I appreciated the findings in the manuscript, I would have liked to see more detail provided in the methods, especially related to how the authors worked to identify the sources of sounds. For example, in the results on lines 63 and 79, the authors state they recorded corixids and a plant in aquaria to supplement their field recordings, but the aquaria testing is not described at all in the methods. They could also include a list of the sources they used for identifying previously documented sound producers and more detail as to how the visual sound-producer assessments were conducted (e.g., were they in the water or on the shore or both, how many people were involved in the assessments). Similarly, I would appreciate if the authors could provide an idea of how many recordings at what lengths and over how many days and locations within the lakes were assessed as part of the study. Because the field of underwater bioacoustics has historically struggled with misidentified sound sources, I believe adding more information like this to the methods would aid readers’ confidence in the authors’ findings.

I had several other recommendations related to the manuscript, outlined below, encompassing the possible inclusion of more acoustic measurement statistics and some writing corrections. I consider all of the revisions recommended relatively minor.

Lines 8, 170, and possibly elsewhere: Sentences shouldn’t begin with an acronym, so the acronyms should be either spelled out or the sentence should be reworded.

Lines 8, 76, 85, and possibly elsewhere: Sago is sometimes capitalized, sometimes not, so this should be made consistent throughout.

Line 46: There is a period missing at the end of this sentence.

Line 88: The authors state in the table caption that they provide “average peak frequency” but then the table values themselves seem to be more reflective of ranges, so perhaps the caption should be corrected. Relatedly, however, if possible, I think it would be the most helpful to provide the mean, standard deviation or error, and ranges of the peak frequencies and durations from each of the exemplar recordings as different readers might prefer some or all of the values for different purposes. Additionally, there is not a widely used standard for referring to individual units of multi-part sounds, here referred to as “syllables.” Because of this, it might be worthwhile to define what is meant by syllable, either in the caption itself or in the methods.

Table 1, Line 111, and possibly elsewhere: When reporting a range of values, an en dash “–“ should be used rather than a hyphen.

Line 98: I really appreciated the authors onomatopoetic descriptions here and elsewhere along with the more quantitative descriptions of the sounds.

Lines 170, 184, and possibly elsewhere: When referring to multiple fish species (rather than multiple fish individuals that are the same species), then the plural of “fish” should be “fishes”.

Line 171: I believe this reference citation should be changed from “Wilson 2004” to “30”.

Line 191: I believe “soundscape” should be made plural to fit the grammatical sentence structure.

Line 213 and elsewhere: The more commonly used name for these types of figures are spectrograms, whereas I believe spectrograph usually refers to something that splits light into its component colors. Also, from the figures, the authors could note here in the methods that they created both spectrograms and waveforms of their exemplar recordings.

References: Though not a necessary inclusion, the authors may find this recent publication of interest: Desjonquères C, Linke S, Greenhalgh J, Rybak F, Sueur J (2024) The potential of acoustic monitoring of aquatic insects for freshwater assessment. Philos Trans R Soc B 379:20230109. Also, in glancing through the references, I did find a few errors, such as the use of title case instead of sentence case in reference 2, listing the journal of reference 31 as AFS instead of Fisheries, and an extra space after the year in line 27, so it may be worth checking them all again during revision.

Reviewer #2: This is an interesting piece of research that quite rightly highlights the importance of sound production by submerged aquatic plants and provides a description of sound production by Stuckenia pectinata. However, I feel that there are some key pieces of information missing from the manuscript that should be added before publication. For example, I think that the introduction could be expanded slightly to include a few key papers (see suggestions below). Also, more detail is required in the methods section in general, and some methods appear in the results section (e.g., aquarium recordings and the use of gas chromatography) but are not described in the methods section. You mention that the recordings were uploaded to Xeno Canto in the results - I would also put this in a Data Availability Statement at the end of the manuscript. I also think that the discussion and conclusion could be improved by thinking about the wider implications for monitoring aquatic plant sounds. These could include the use of aquatic plant sounds as indicators of pollution events / measuring the primary productivity of an ecosystem. Also, many macrophytes are indicator species and are used to assess the ecological condition of an ecosystem. Perhaps the ability to identify key macrophyte species’ sounds will allow for the assessment of freshwater ecosystems using passive acoustic monitoring?

Reviewer #3: This study provides field characterization of an underappreciated component of freshwater soundscapes. I believe this should be published, as it will help progress efforts in freshwater bioacoustic monitoring. However, currently there is not enough detail in the methods and results to determine the initial goal of the study, how the soundscape was sampled, or back up the claim that bubble streams from macrophytes are a dominant part of the soundscape.

Introduction:

Line 21: I see what you’re saying here, but the last two sentences of the first introduction paragraph initially seem to be contradictory. Simply swapping the order of sentences would make your point clearer and improve the flow into the next paragraph.

Line 46: It is worth acknowledging that most sounds attenuate do not propagate well in shallow water, so this is not necessarily a characteristic of these specific signals. Soundscapes may be highly localized, but specifically for shallow-water organisms.

Relevant citations:

• Forrest, T. G., G. L. Miller, and J. R. Zagar. 1993. Sound propagation in shallow water: Implications for acoustic communication by aquatic animals. Bioacoustics 4:259–270.

• Fine, M. L., and M. L. Lenhardt. 1983. Shallow-water propagation of the toadfish mating call. Comparative Biochemistry and Physiology -- Part A: Physiology 76:225–231.

Methods:

There should be much more detail here. It is not clear whether you were passively looking at the whole soundscape (stationary hydrophone with paired direct or camera-based observation through duration of recording) or actively characterizing possible components of the soundscape (selecting possible sound sources based on visual observation and placing a hydrophone near them).

There is almost no discussion of the analysis of the recordings, although it is discussed in the results. How were the visual assessments linked to the analysis of recordings? Were all recordings reviewed manually?

Line 205: How many cumulative minutes of recording did you obtain? What was the distribution of times of day? If you were actively seeking specific sound sources, please list which possible sound sources were targeted and how many recordings were taken of each.

Line 207: Was the hydrophone stationary or mobile? My impression is that you selected potential sound sources and placed the hydrophone to record potential sounds (in the results, you mention following a school of fish), but that is not clear here. Please also note approximate position of the hydrophone in the water column- obviously most of your recording regions were very shallow, but it would be nice to know whether the hydrophone depth was at surface, bottom, or varied to match the sound sources. If you recorded specific sound sources, please note how many recordings were taken for each potential sound source, and possibly the variation in time of day of recording (especially for macrophytes!). Please also note the sample rate of the hydrophone.

Line 209: More detail needed on visual assessments as well. Were both cameras and direct observation used for each recording? Were all possible sound sources noted for each recording? How exactly were the visual assessments and recordings linked?

Line 213: All uses of “spectrograph” should be replaced with “spectrogram”

Line 216: How were the peak frequencies and duration of repeated sections characterized?

Line 218: Unclear what “additional recordings were also considered” means, especially since I do not see them in the results. If they were analyzed for soundscape components, please describe the methods of recording and analysis. If they were not analyzed in a way that could be used in the results, they should not be in the methods.

Results

As noted, there was no discussion of the analysis in the methods that yielded the first paragraph of the results. As such, it is difficult to interpret statements like the first sentence- were they simply the only sound source in most of the recordings, or were they stronger in volume? Were these sounds found in most recordings? It would be informative to have some more numbers here. If the hydrophones were placed specific to a given potential sound source, what percentage of the recordings actually yielded a distinctive signal? If the hydrophones were placed randomly to capture an entire soundscape, what percentage of recordings featured each type of sound, and how did the occurrences of a visually observed sound source align with the occurrences of its associated sound? When multiple sound sources were present, which was higher in volume?

I would recommend some restructuring of this section to first enumerate what species were present, then how often each produced sounds.

Line 57: Are you certain these are methane bubbles?

Line 60: Were boats present during recordings? As above, some description of the comparisons between your visual and auditory surveys would contribute greatly to this section.

Line 63: Aquaria recordings were not described in the methods.

Line 64: Were abundances of insects anecdotally observed or quantified in any way? If there were more potential sound makers than were notably absent from the recordings, please include all of their names.

Line 73: If there were more fish species observed that did not make sounds, please list them; if not, change wording of sentence to reflect that these are all fish that were observed in this study.

Line 79: How was the aquarium plant “roughly comparable” to Exemplar 3? Similar in frequency distribution, similar in pulse rate?

Line 85: Inconsistent capitalization of Sago/sago.

Line 95: Since the analysis was not described in the methods, I can’t tell how all of these descriptive measures were decided. Did you measure each syllable individually? If so, then “duration of syllables” in the table should be mean duration of syllables, and it would be nice to include a measure of spread (standard deviation, etc.). This also seems like the minimum and maximum peak frequencies from the spread across each syllable, but it’s hard to determine.

Line 122: Unless part of the signal that was analyzed is not in the figure, I don’t see what part of this signal would likely have had a peak frequency of 6.2 kHz.

Line 147: Should be Exemplar 4

Discussion

Line 155: I do not see enough evidence in the results to support this claim. Please back this up with how the prevalence of the bubble streams compared to other signals. There was also no further discussion of the corixid stridulations beyond the first sentence of the results section. Even if the corixid stridulation has been described in other papers, it would be beneficial to include a description of the characteristics of the stridulation signal in this setting.

Line 158: Fully submerged plants typically do not have stomata, and I believe this is the case for sago pondweed. This does not negate the probability of these bubble streams being from wounds, but any “pearling” would not be associated with stomata on these plants.

Line 164: Acoustic ecologists are increasingly using machine learning algorithms to increase the efficiency of reviewing acoustic recordings. Can you describe any commonalities of the bubble streams that may be helpful for programming automated detectors? Additionally, you noted that Exemplar 4 was fairly distinct among the sounds. Were there characteristics of the plant or bubble stream that might explain this?

Line 169: There are many other sound sources and types noted across all aquatic habitats- marine invertebrates, tonal fish calls, etc. This sentence is likely more accurate with respect to freshwater habitats.

Line 171: Citation formatting

Line 188: Good points here

Line 192: I think it would be worth acknowledging that all the bubble stream signals in this paper are from the same type of plant. Is this plant sufficiently ubiquitous in lakes in Canada that this is widely relevant? Are these bubble streams likely to be similar among freshwater species? What is the depth limit of Sago pondweed?

Figures:

The font is small and a bit grainy- is there a way to get higher resolution, at least on the text? An image of a bubble stream might also make a good addition to your figures, especially since you make a distinction between a bubble stream from a wound and "pearling".

6. PLOS authors have the option to publish the peer review history of their article (what does this mean? ). If published, this will include your full peer review and any attached files.

**Do you want your identity to be public for this peer review?** For information about this choice, including consent withdrawal, please see our Privacy Policy .

Reviewer #1: No

Reviewer #2: **Yes: ** Jack Greenhalgh

Reviewer #3: No

---

## [Author Response · Author response to Decision Letter 1]

12 Dec 2024

Response to all reviewer comments have been indluced in the attachment "Response to Reviewers.docx"

---

## [Editor Report · Decision Letter 1]

29 Dec 2024

Photosynthetic oxygen bubble stream sounds from aquatic macrophytes, and their consequences for acoustic biodiversity inventories and acoustic communication in shallow freshwater settings

PONE-D-24-33566R1

Dear Dr. Campbell,

We’re pleased to inform you that your manuscript has been judged scientifically suitable for publication and will be formally accepted for publication once it meets all outstanding technical requirements.

Kind regards,

Charles William Martin

Academic Editor

PLOS ONE
---

## [Editor Report · Acceptance letter]

PONE-D-24-33566R1

PLOS ONE

Dear Dr. Campbell,

I'm pleased to inform you that your manuscript has been deemed suitable for publication in PLOS ONE. Congratulations! Your manuscript is now being handed over to our production team.

Kind regards,

on behalf of

Dr. Charles William Martin

Academic Editor

PLOS ONE